# Novel neuroanatomical integration and scaling define avian brain shape evolution and development

Akinobu Watanabe[1,2,3]*, Amy M Balanoff[2,4], Paul M Gignac[2,5], M Eugenia L Gold[2,6], Mark A Norell[2]

[1]Department of Anatomy, New York Institute of Technology College of Osteopathic Medicine, Old Westbury, United States; [2]Division of Paleontology, American Museum of Natural History, New York, United States; [3]Department of Life Sciences Vertebrates Division, Natural History Museum, London, United Kingdom; [4]Department of Psychological and Brain Sciences, Johns Hopkins University, Baltimore, United States; [5]Department of Anatomy and Cell Biology, Oklahoma State University Center for Health Sciences, Tulsa, United States; [6]Biology Department, Suffolk University, Boston, United States

*For correspondence:
awatanab@nyit.edu

Competing interests: The authors declare that no competing interests exist.

**Abstract** How do large and unique brains evolve? Historically, comparative neuroanatomical studies have attributed the evolutionary genesis of highly encephalized brains to deviations along, as well as from, conserved scaling relationships among brain regions. However, the relative contributions of these concerted (integrated) and mosaic (modular) processes as drivers of brain evolution remain unclear, especially in non-mammalian groups. While proportional brain sizes have been the predominant metric used to characterize brain morphology to date, we perform a high-density geometric morphometric analysis on the encephalized brains of crown birds (Neornithes or Aves) compared to their stem taxa—the non-avialan coelurosaurian dinosaurs and *Archaeopteryx*. When analyzed together with developmental neuroanatomical data of model archosaurs (*Gallus*, *Alligator*), crown birds exhibit a distinct allometric relationship that dictates their brain evolution and development. Furthermore, analyses by neuroanatomical regions reveal that the acquisition of this derived shape-to-size scaling relationship occurred in a mosaic pattern, where the avian-grade optic lobe and cerebellum evolved first among non-avialan dinosaurs, followed by major changes to the evolutionary and developmental dynamics of cerebrum shape after the origin of Avialae. Notably, the brain of crown birds is a more integrated structure than non-avialan archosaurs, implying that diversification of brain morphologies within Neornithes proceeded in a more coordinated manner, perhaps due to spatial constraints and abbreviated growth period. Collectively, these patterns demonstrate a plurality in evolutionary processes that generate encephalized brains in archosaurs and across vertebrates.

## Introduction

The human brain, with its inflated cerebrum, is often considered the zenith of brain evolution. Seminal works, both classic and modern, have suggested that our specialized brain morphology arose through (i) changes in gross-level scaling relationship (allometry) of brains (*Striedter, 2005*; *Rilling, 2006*; *Passingham and Smaers, 2014*), and (ii) mosaic, or modular, evolution where individual brain regions have the capacity to evolve quasi-independently from one another due to decoupling of previously shared genetic, developmental, functional, and spatial constraints (*Barton and Harvey, 2000*; *Rowe et al., 2011*; *Smaers and Soligo, 2013*; *Gómez-Robles et al., 2014*; *Ni et al., 2019*). Clarifying the degree to which these patterns extend across vertebrates requires examining other

episodes of encephalization. Crown birds offer an excellent comparative system to mammals, even primates, because they share neuroanatomical features that evolved independently, including a relatively large brain size (*Jerison, 1973*; *Nieuwenhuys et al., 1998*; *Northcutt, 2002*; *Butler and Hodos, 2005*; *Iwaniuk et al., 2005*; *Gill, 2006*), globular brains with expanded cerebra, specialized cytoarchitecture and neuron types (*Reiner et al., 2004*; *Dugas-Ford et al., 2012*; *Shanahan et al., 2013*; *Pfenning et al., 2014*; *Karten, 2015*; *Stacho et al., 2020*), and the capacity to perform higher cognitive behaviors (*Lefebvre et al., 2002*; *Weir et al., 2002*; *Emery, 2006*; *Auersperg et al., 2012*; *Kabadayi et al., 2016*; *Bayern et al., 2018*; *Boeckle et al., 2020*). In addition, they feature remarkable variation in brain morphology that is conducive to macroevolutionary studies (*Iwaniuk and Hurd, 2005*; *Figure 1*).

Chronicling the evolutionary origins of the archetypal 'avian' brain requires information on ancestral brain morphologies of extinct coelurosaurian dinosaurs. Because brain tissue does not readily fossilize, paleontologists have relied on endocasts, or the internal mold of the braincase, to document and analyze neuroanatomical evolution through geologic time (*Jerison, 1963*; *Jerison, 1969*; *Edinger, 1975*; *Hopson, 1979*; *Balanoff and Bever, 2017*). As in extant mammals, the brain occupies nearly the entire cranial cavity in crown birds, and thus, these endocasts are used as accurate proxies for brain size and shape in these groups (*Jerison, 1973*; *Haight and Nelson, 1987*; *De Miguel and Henneberg, 1998*; *Iwaniuk and Nelson, 2002*; *Watanabe et al., 2019*; *Early et al., 2020*). Volumetric analyses of endocasts from avialan and non-avialan dinosaurs show that crown birds exhibit a derived allometric trend in brain-to-body size although some closely related non-avialan dinosaurs (e.g., oviraptorosaurs, troodontids) overlap in allometric trends with neornithine groups (*Balanoff et al., 2013*; *Ksepka et al., 2020*). Volumetric data of endocasts also indicate that each brain region evolved under different modes across avian and non-avian coelurosaurs, implying mosaic brain evolution (*Balanoff et al., 2016b*). However, whether the encephalized brains of crown birds possess a unique allometric trajectory and a more modular structure than non-avialan archosaurs, as anticipated by classic notion of phenotypic modularity (*Wagner and Altenberg, 1996*; *Goswami et al., 2014*), remains to be explicitly tested. In addition, despite its prevalence and importance as a morphological metric, size data are limited in characterizing brain morphology. For example, similarly sized brains could have disparate shapes, especially given the diversity in brain morphologies within Neornithes and similar volumetric proportions between crown birds and some non-avialan dinosaurs.

To holistically analyze neuroanatomical shape, we use a high-density geometric morphometric (GM) approach on endocranial reconstructions from micro-computed tomography (µCT) imaging. Three-dimensional (3D) landmarks were placed virtually on endocasts from 37 extant and recently extinct (Dodo, Greak Auk) neornithine species and six non-avialan coelurosaurs to characterize the overall morphology of the brain and its functional subdivisions visible on the endocast—cerebrum, optic lobe, cerebellum, and medulla (*Figure 1*; *Supplementary file 1a, b* for specimen and landmark sampling). In this study, endocranial regions are referred to by the name of the soft-tissue features that are reflected on the surface. This unified mathematical framework allows the relative size, configuration, and surface morphology of neuroanatomical traits to be analyzed together, including allometric and correlative trends in shape within and between brain regions (*Klingenberg, 2008*). We analyze this rich phenotypic dataset to test if crown birds exhibit (i) a derived allometric relationship between endocranial shape and size, and (ii) a more structurally modular brain compared to the ancestral pattern observed in non-avialan coelurosaurian dinosaurs. Moreover, we anticipate that differences in evolutionary patterns of allometry and phenotypic integration will be reflected in extant archosaurs developmentally (*Bookstein et al., 2003*), where clades with more integrated brain evolution show more integrated brain development. As such, we combine postnatal developmental data of the American alligator (*Alligator mississippiensis*) and the domestic chicken (*Gallus gallus*) with interspecific sampling of Coelurosauria to assess whether (iii) the developmental allometry and integration pattern mirror evolutionary patterns of endocranial shapes in non-avialan dinosaurs and crown birds, respectively.

## Results

Shape data were subjected to principal components analysis (PCA) to create morphospaces that visualize patterns of neuroanatomical variation (*Figure 2*; *Figure 2—figure supplement 1* for fully

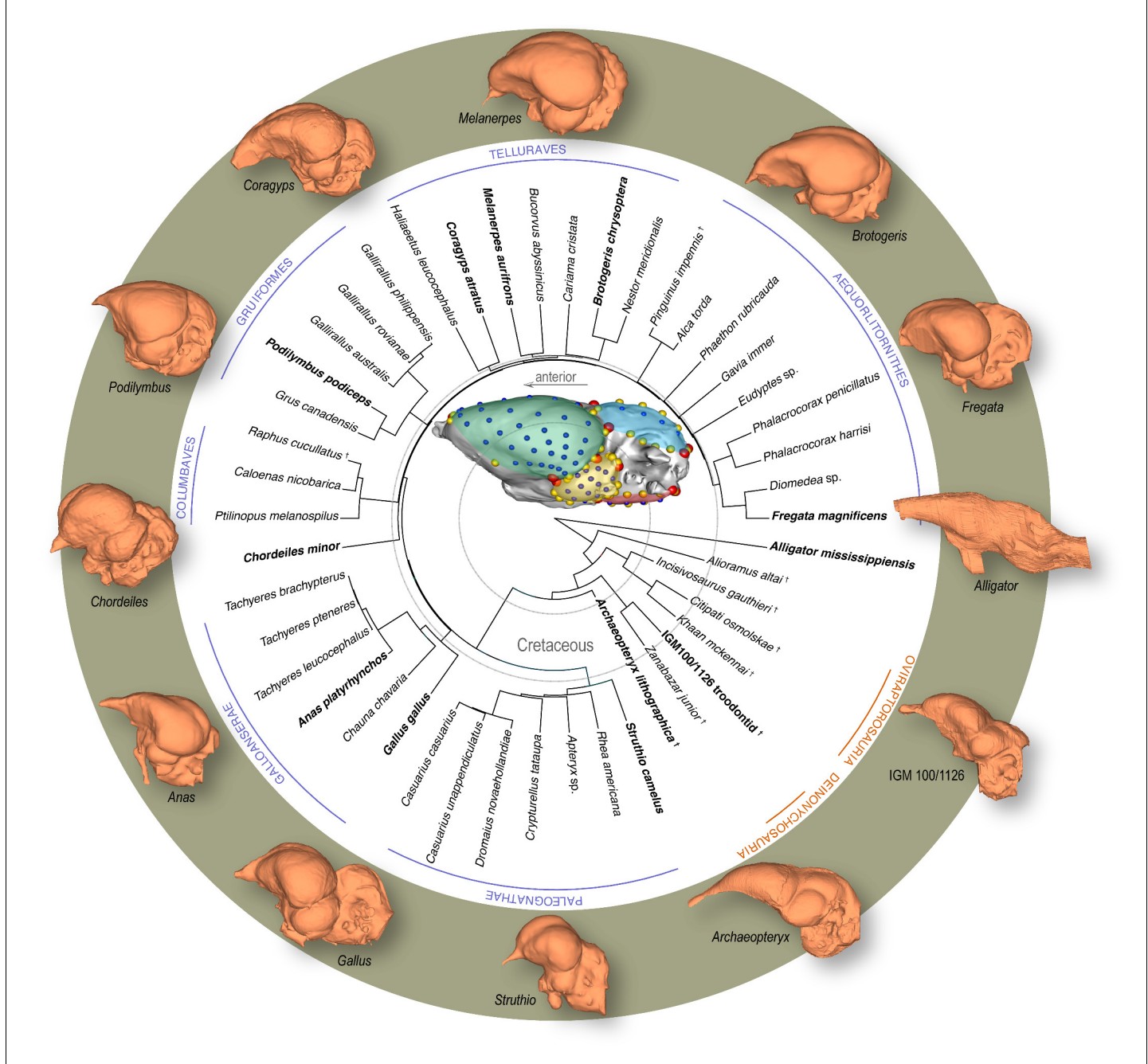

**Figure 1.** Time-calibrated phylogeny of avialan and non-avialan coelurosaurs sampled in this study, with *Alligator mississippiensis* as outgroup. Center image shows discrete (red), curve (yellow), and surface (blue) landmarks used to characterize endocranial shape including the cerebrum (green), optic lobe (yellow), cerebellum (blue), and medulla (red). Lateral views of select endocranial models, indicated by bolded taxonomic names on the phylogeny, highlight the neuroanatomical variation observed across taxa. See *Supplementary file 1a* for list of specimens sampled for the interspecific dataset and *Supplementary file 1b* for the landmark scheme used in this study.

labeled morphospaces). Morphospace of overall endocranial shape shows that Neornithes exhibits distinct brain morphologies from non-avialan archosaurs (multivariate analysis of variance: $R^2$ = 0.323; p<0.001), largely along the PC1 axis (*Figure 2a*). *Archaeopteryx*, often considered one of the earliest diverging avialans (*Pittman et al., 2020*), and an unnamed troondontid (IGM 100/1126) occupy an intermediate position between non-avialan dinosaurs and Neornithes, indicating general evolutionary trend toward the neornithine brain form as previously reported (*Balanoff et al.,*

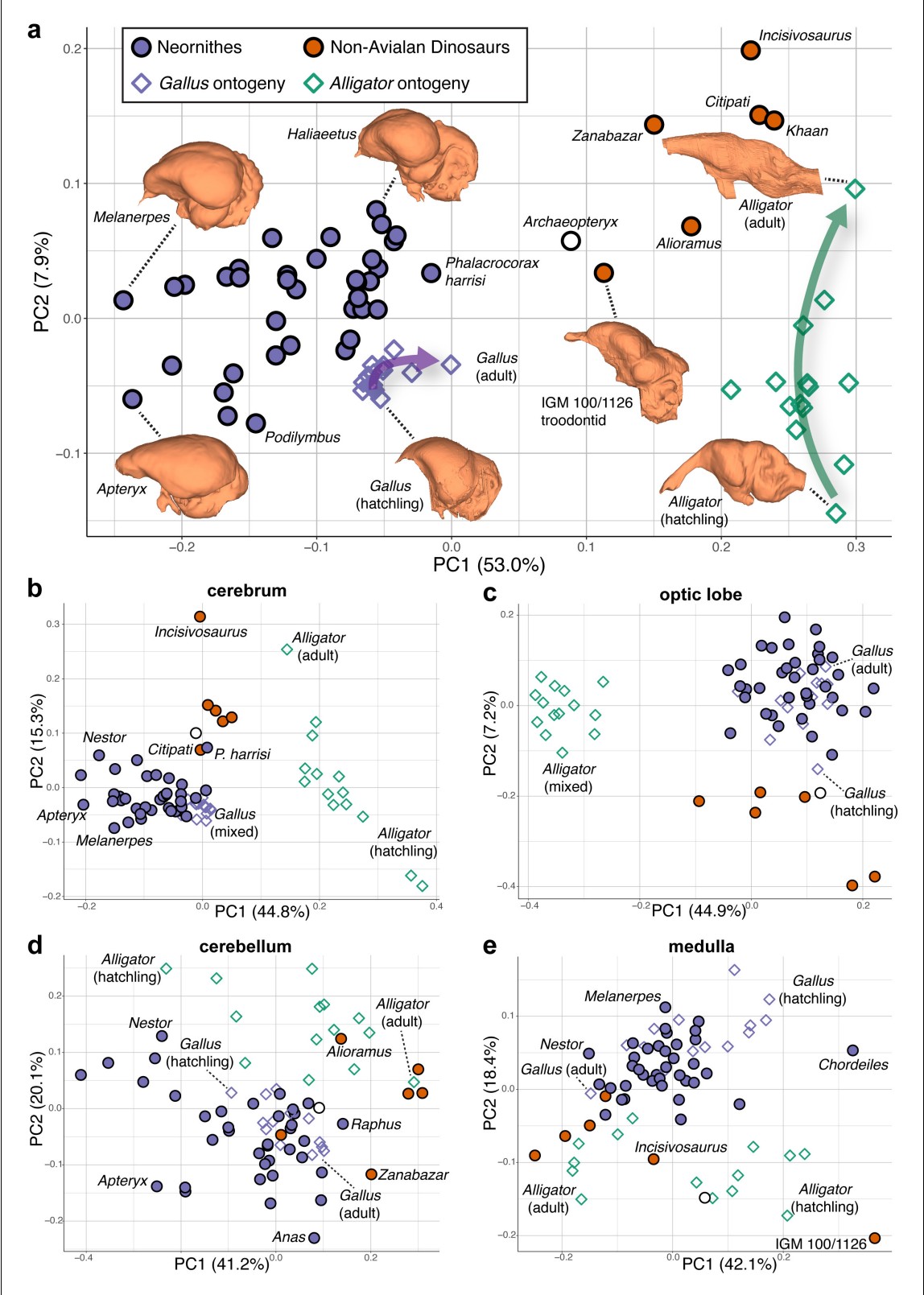

**Figure 2.** Morphospaces constructed from first two principal components (PC) of neuroanatomical shapes. These plots illustrate the distribution of shape variation in the (a) overall endocranial shape, where the arrows denote postnatal developmental trajectories of *Alligator* (green) and *Gallus* (purple); (b) cerebrum; (c) optic lobe; (d) cerebellum; and (e) medulla. Regional shape data are locally aligned. See text for details. The following figure supplement is available for *Figure 2—figure supplement 1*. PC morphospaces with full specimen labels.

*Figure 2 continued on next page*

*Figure 2 continued*

The online version of this article includes the following figure supplement(s) for figure 2:

**Figure supplement 1.** Morphospaces constructed from first two principal components of neuroanatomical shape.

*2014*; *Ksepka et al., 2020*). Although size data from endocasts show partial overlap of crown birds and non-avialan coelurosaurs (*Balanoff et al., 2013*; *Ksepka et al., 2020*), high-density shape data discriminate these groups more clearly along PC1 axis, where lower PC1 scores in the morphospace (*Figure 2a*) are associated with expanded cerebrum, ventrally located optic lobe, more compact hindbrain, and greater dorsoventral flexion. Lower PC2 scores correlate with wider cerebrum, dorso-ventrally longer optic lobe, anteroposteriorly shorter cerebellum, and more dorsoventrally flexed medulla. Besides the distinction between non-avialan dinosaurs and crown birds, the distribution of endocranial shape variation within Neornithes has a broad, but modest phylogenetic structure with substantial overlap and convergence among subclades (Blomberg's $K$ = 0.035; p=0.039; see *Supplementary file 1c* for phylogenetic signal in shape data). The developmental trajectory of *Alligator* occupies the area of morphospace farthest from crown birds, whereas the endocranial shapes of developing *Gallus* lie adjacent to the cluster of crown birds. When morphospaces are constructed for locally aligned shape data of individual brain regions, cerebrum and optic lobe shapes largely separate *Alligator*, non-avialan coelurosaurs, and crown birds (*Figure 2b,c*), whereas cerebellum and medulla shapes partially overlap between these major clades (*Figure 2d,e*).

Upon establishing that non-avialan coelurosaurs and crown birds diverge in overall endocranial shape, we examined whether this difference is associated with deviations in their scaling relationships. After correcting for phylogenetic structure in the data, endocranial size, as measured by log-transformed centroid size, accounts for 24.2% and 4.8% of total endocranial shape variation within non-avialan coelurosaurs and crown birds, respectively (albeit p>0.05; see *Supplementary file 1c* for allometric signal in shape data). Collectively, these values indicate that size captures a relatively small proportion of neuroanatomical variation, especially in crown birds as previously reported (*Marugán-Lobón et al., 2016*). To visualize how each endocast diverges from the overall allometric trend across all sampled endocasts, we created bivariate plots of PC1 of residuals from the common allo-metric trend (RSC1) against scores along this allometric trendline (common allometric component [CAC]) where increase in its value corresponds to increase in size (*Mitteroecker et al., 2004*; *Figure 3*; *Figure 3—figure supplement 1* for fully labeled plots). The plot for overall endocra-nial shape illustrates that endocranial shape variation of non-avialan dinosaurs and crown birds lies along divergent allometric trajectories (*Figure 3a*; non-avialan coelurosaur-Neornithes difference in allometric trajectories: $R^2$ = 0.273; p<0.001). Still, *Archaeopteryx* is positioned between Neornithes and other non-avialan archosaurs along the shape axis and allometric trajectories (*Figure 3a*). Thus, for its size, the endocranial shape of *Archaeopteryx* at the nexus of the theropod-bird transition exhibits an intermediary form that approaches that of crown birds. When the developmental dataset of *Alligator* and *Gallus* are incorporated into the interspecific data, neuroanatomical changes in developing *Gallus* overlie the interspecific allometric trajectories of crown birds, whereas those in *Alligator* more closely match the interspecific allometric trajectories of non-avialan coelurosaurs (*Figure 3a*). Multivariate analysis of variance in full shape space rejects the null hypothesis that the allometric trajectories are shared between developing *Alligator* and *Gallus* ($R^2$ = 0.609, p<0.001 respectively). These results clearly indicate that the avian crown possesses a derived brain-to-size relationship that governs their brain shape evolution as well as development.

Next, to investigate whether evolutionary shifts in allometric trajectories occurred in a concerted or mosaic pattern across regions, we compared allometric trajectories for individual brain regions. For all brain regions, endocranial shapes of *Gallus* do not overlap with those of *Alligator* throughout their postnatal development (*Figure 3b–e*). The degree to which non-avialan coelurosaurs deviate from the allometric trajectories of endocranial shape in *Alligator* and Neornithes varies for each region. In cerebrum shape, the non-avialan coelurosaurs span the intermediate space between *Alli-gator* and Neornithes (along PC1 residual shape score), with *Archaeopteryx* and the oviraptorid *Citi-pati* within the cluster of crown birds (*Figure 3b*). Based on optic lobe shape, non-avialan coelurosaurs are isolated from the allometric trend of *Alligator* but display closer affinity to crown birds and *Archaeopteryx* (*Figure 3c*). With the cerebellum, the troodontids (*Zanabazar* and IGM

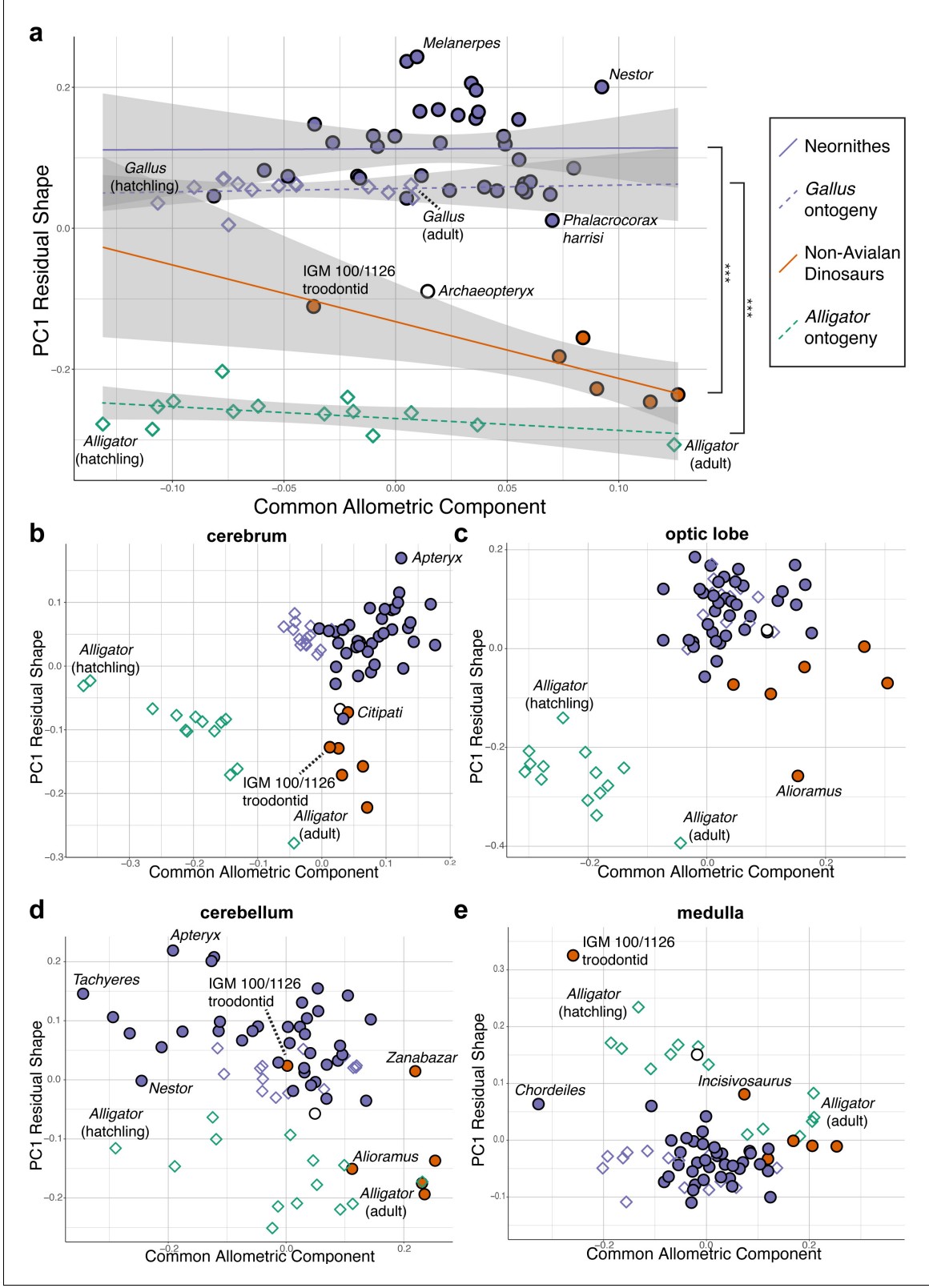

**Figure 3.** Bivariate plots of PC1 of residuals from the common allometric component (CAC) against scores along CAC (*Mitteroecker et al., 2004*). These plots illustrate neuroanatomical deviations from the overall shape-to-size allometric trend in the (**a**) endocasts (band indicates 95% confidence band), where the null hypothesis that the allometric trajectories between Neornithes and non-avialan dinosaurs and between *Alligator* and *Gallus* are the same is rejected statistically (*** denotes p<0.001); (**b**) cerebrum; (**c**) optic lobe; (**d**) cerebellum; and (**e**) medulla. For each subregion, locally aligned

*Figure 3 continued on next page*

*Figure 3 continued*

shapes and regional log-transformed centroid sizes were used. See text for details. The following figure supplement is available for *Figure 3—figure supplement 1*. Principal components (PC) plots of PC1 of residuals from CAC against CAC with full specimen labels.

The online version of this article includes the following figure supplement(s) for figure 3:

**Figure supplement 1.** Bivariate plots of PC1 of residuals from the common allometric component (CAC) against scores along CAC.

100/1126) follow the allometric trajectory of crown birds, whereas the other non-avialan coelurosaurs align with the developmental trajectory of *Alligator* (*Figure 3d*). Developmental trajectories of medulla shape are distinct between *Alligator* and *Gallus* but converge as individuals of these taxa grow (*Figure 3e*). The troodontid IGM 100/1126 and *Incisivosaurus* exhibit medulla shapes that are more consistent with allometric trends in *Alligator* development, whereas the correspondence of other non-avialan coelurosaurs to the allometric trajectories of medulla shape in *Alligator* or crown birds are ambiguous due to the convergent allometric trajectory at these medulla sizes (*Figure 3e*).

Lastly, we employed two methods for evaluating the pattern of integration—covariance ratio (CR) (*Adams, 2016*) and maximum likelihood (ML) (*Goswami and Finarelli, 2016*) approaches—to calculate and test the strength of correlation between shapes of neuroanatomical regions. Results from these analyses would elucidate whether the derived neornithine allometric trajectory accompanied a shift in the pattern of morphological integration in the brain. We find that non-avialan coelurosaurs and crown birds reveal different patterns of integration (*Figure 4*; *Supplementary file 1d–f* for within- and between-region correlation values). Both CR and ML analyses indicate that non-avialan coelurosaurs show strong associations between the cerebrum and optic lobe and between the medulla and cerebellum, while only the CR analysis indicates stronger correlation between the cerebrum and medulla. The results for neornithines show contrasting patterns between analyses, where CR analysis suggests strong integration between the optic lobe and medulla and ML analysis presents a strong correlation between the cerebrum and cerebellum. Despite this discrepancy, comparison of correlation values clearly indicates that integration between brain regions is stronger in crown birds than in non-avialan dinosaurs (*Supplementary file 1d–f*). In particular, crown birds possess a cerebellum that is much more strongly integrated with the cerebrum and optic lobe shapes, which are pairs of structures with much weaker correlations in non-avialan dinosaurs.

Developing *Alligator* and *Gallus* also show contrasting patterns of integration. *Alligator* shows the strongest integration between the optic lobe and medulla, with slightly weaker correlation between the cerebrum and optic lobe. In contrast, *Gallus* shows strong integration between the cerebellum and medulla and more moderate correlations between the cerebrum and optic lobe and between the cerebrum and cerebellum. As observed in non-avialan dinosaurs, cerebellum shape in *Alligator* is weakly correlated with cerebrum and optic lobe shapes, whereas these associations are much stronger in developing *Gallus* akin to the evolutionary pattern seen in crown birds. When tested for differences in the effect size of integration, we find that the endocranial shape of crown birds exhibits significantly greater integrated structure than non-avialan dinosaurs ($p < 0.001$ for overall and pairwise regions; *Supplementary file 1g*). While statistical significance is lacking for most comparisons in neuroanatomical integration between developing *Alligator* and *Gallus*, the pairwise tests with moderate statistical significance ($p < 0.05$) show that *Gallus* undergoes more integrated shape changes between the cerebrum and cerebellum and between cerebellum and medulla (*Supplementary file 1g*).

## Discussion

### Neornithes exhibits derived brain shape, allometry, and integration pattern

Our results indicate that crown birds (i) follow a distinct brain shape-to-size scaling relationship and (ii) possess a more integrated brain structure compared to non-avialan archosaurs that (iii) uniquely characterizes their brain development and evolution. While these derived features of a neornithine brain are clearly demonstrated here, the evolutionary origin of these neuroanatomical novelties is complex. First, the proximity of non-avialan coelurosaurs to crown birds in endocranial shape and its allometric trajectory varies across neuroanatomical regions. The shape differences are more

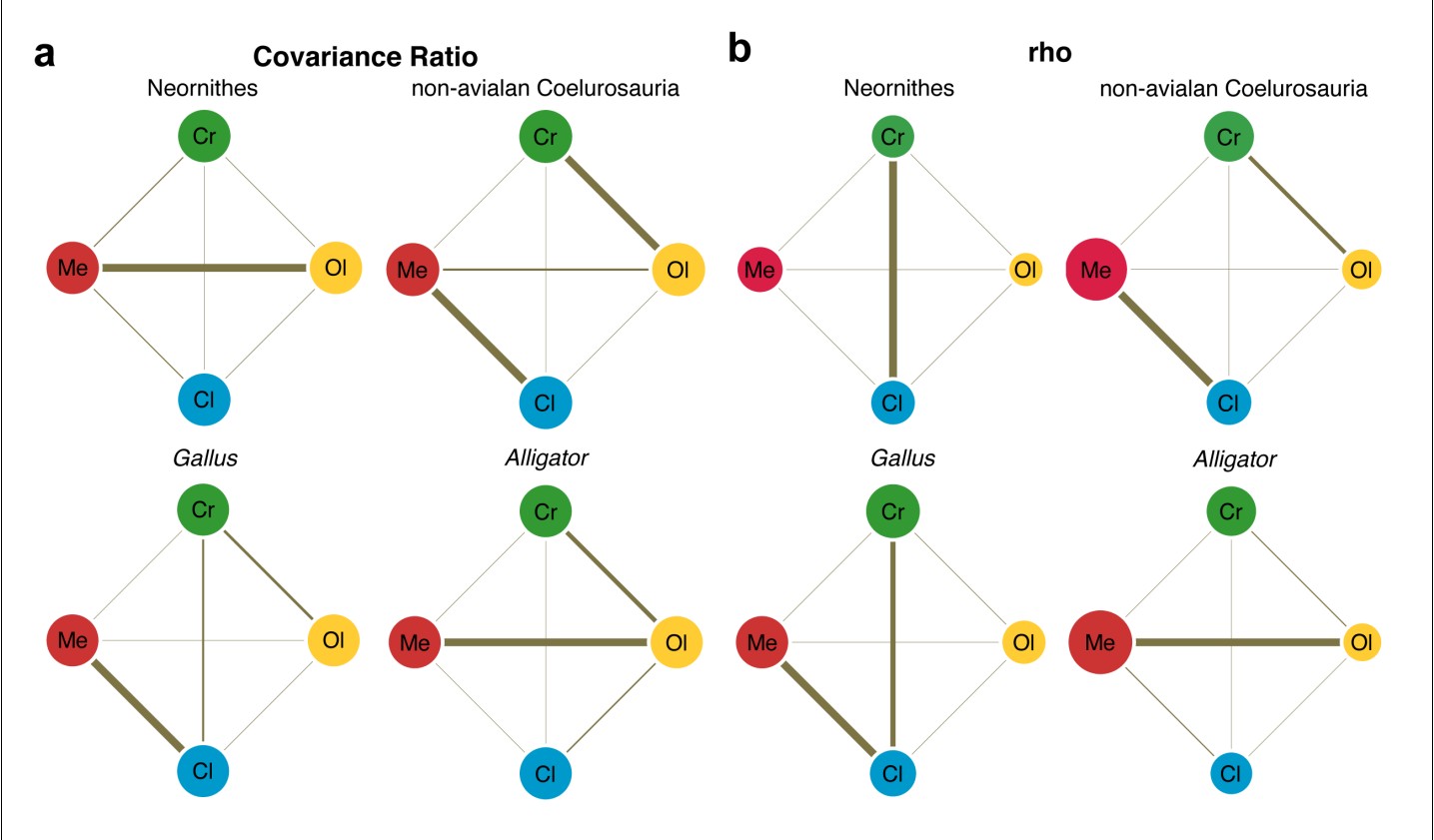

**Figure 4.** Pattern of correlation across locally aligned neuroanatomical shapes. (**a**) Network diagrams based on between-region covariance ratios (CR) (*Supplementary file 1d, f*; *Adams, 2016*). (**b**) Network diagrams based on correlation coefficient, rho, from maximum likelihood analysis (*Supplementary file 1d, f*; *Goswami and Finarelli, 2016*), where the size of the circles represent the degree of within-region correlation. In both sets of diagrams, the thickness of the line segments between regions indicates relative strength of the correlation. Note that the line thickness is based on values within each analysis (i.e., not comparable between diagrams), where the cut-off point is the mean correlation value. Abbreviations: Cl, cerebellum; Cr, cerebrum; Ol, optic lobe; Me, medulla. See text for details. The following figure supplement is available for *Figure 4—figure supplement 1*. Network diagrams of integration within and between globally aligned neuroanatomical regions.

The online version of this article includes the following figure supplement(s) for figure 4:

**Figure supplement 1.** Pattern of correlation across globally aligned neuroanatomical shapes.

pronounced in the cerebrum and optic lobe (*Figure 3b,c*) than in the hindbrain (*Figure 3d,e*). *Archaeopteryx* shows closer resemblance to non-avialan coelurosaurs than extant birds in cerebrum, optic lobe, and medulla shapes, demonstrating that the organization of the archetypal 'avian' brain had not emerged by the origin of Avialae, if *Archaeopteryx* is considered one of the earliest diverging avialans (*Figure 3b–d*). However, allometric trends across regions signify that *Archaeopteryx* possessed an avian-grade shape-to-size relationship for the optic lobe, and nearly so for the cerebrum and cerebellum. In fact, some non-avialan coelurosaurs approach or align with the allometric trends of crown birds, including non-avialan maniraptoran dinosaurs for the optic lobe and troodontids in cerebellum shape. The oviraptorosaur *Citipati* converges on the cerebral shape of *Archaeopteryx* and the extant cormorant *Phalacrocorax* given its size (*Figure 3b*). These taxon- and region-specific results are consistent with volumetric studies reporting a mosaic assembly of the avian brain form (*Balanoff et al., 2016b*; *Ksepka et al., 2020*). Based on our endocranial shape data, we propose that optic lobes approaching avian-grade scaling relationships emerged at least among Pennaraptora; and similarly, avian-grade allometric trend in cerebellum shape first appeared among Paraves prior to the origin of Avialae (possibly convergently). These neuroanatomical innovations were then followed by the acquisition of specialized shape-to-size scaling relationships in the cerebrum along the lineage spanning *Archaeopteryx* to crown birds, potentially coincident with the

increased prominence of the Wulst, a dorsal telencephalic eminence that receives somatosensory and visual signals and thought to be involved in information processing and motor control implicated in powered flight (*Gold et al., 2016*). Notably, this inferred change to cerebrum shape and development is decoupled from cerebrum size evolution which did not change substantially from non-avialan paravians to crown birds (*Balanoff et al., 2013*; *Ksepka et al., 2020*).

A difficulty in pinpointing the evolutionary origin of derived allometric and integration patterns is the dearth of complete endocranial material from early avialan taxa (*Walsh and Milner, 2011*; *Knoll and Kawabe, 2020*) and robust developmental series of non-avialan coelurosaurian dinosaurs. This gap in neuroanatomical sampling limits our current ability to precisely determine the timing and tempo of when these neuroanatomical properties evolved. While shape data could not be collected, known braincase and endocast from the purported ornithuran *Cerebavis* from the Cretaceous period features an amalgam of neornithine neuroanatomical features, with globular and laterally expanded cerebra, ventrally positioned optic lobes, anteroposteriorly short and ventrally positioned cerebellum, and strongly flexed profile (although without a well-developed Wulst) (*Kurochkin et al., 2007*; *Walsh and Milner, 2011*; *Walsh et al., 2016*; *Beyrand et al., 2019*). Based on these observations, the final phases of the acquisition of an avian-grade brain, including the inflated appearance of cerebra, likely occurred along the lineage spanning the divergence of earliest avialans (e.g., *Archaeopteryx*) to the origin of Ornithurae. Future discoveries of exceptionally preserved braincases of Mesozoic stem birds and their inclusion into our endocranial shape data will greatly facilitate our understanding of this key period in amniote brain evolution. Recent discoveries and reconstructions of complete, articulated, and three-dimensionally preserved Mesozoic avialans (*Field et al., 2018*; *Field et al., 2020*), as well as a developmental series of non-avialan dinosaurs (*Evans et al., 2009*; *Lautenschlager and Hübner, 2013*; *Bullar et al., 2019*), provide a promising outlook on comparative studies of brain evolution along the theropod-bird transition. Regardless of a punctuated or gradual evolution of avian-grade cerebrum among avialans, our study demonstrates that the brain of the avian crown exhibits a distinct allometric trajectory and a more integrated structure as compared to the ancestral, non-avialan theropod condition.

## Developmental trends in coelurosaurian brain evolution

Comparative neuroanatomists have long recognized the intimate connection between brain development and evolution. For instance, prolonged periods of neurogenesis (*Allman et al., 1993*; *Jones and MacLarnon, 2004*; *Leigh, 2004*; *Barton and Capellini, 2011*; *Sayol et al., 2016b*; *Yu et al., 2018*; *Gunz et al., 2020*) and regions with delayed onset of neurogenesis (*Finlay and Darlington, 1995*; *Finlay et al., 2001*; *Charvet and Striedter, 2011*) have been shown to be associated with greater encephalization across vertebrates. In this study, we analyzed interspecific and developmental data synchronously, allowing inference of evolutionary shifts in developmental sequence and rate of phenotypic changes, termed heterochrony (*Gould, 1977*; *Alberch et al., 1979*; *McKinney and McNamara, 1991*; *Klingenberg, 1998*). Paedomorphosis (retention of ancestrally juvenile brain morphology in adult stages of descendants) through early cessation of development, or progenesis, has been proposed to account for the dorsoventrally flexed brain profile observed in crown birds that began among non-avialan paravian dinosaurs (*Beyrand et al., 2019*). This mechanistic explanation agrees with the accelerated yet abridged growth period in birds. The same heterochronic process has been invoked for the evolution of the avian skull (*Bhullar et al., 2012*)—a structure topologically, functionally, and developmentally linked to the brain (*Young et al., 2010*; *Gondré-Lewis et al., 2015*; *Hu et al., 2015*).

Based on our landmark and developmental sampling, we find no evidence of a uniform heterochronic process underlying the evolution of overall endocranial shape from a crocodylian outgroup to crown birds. Both the morphospace and allometric trajectories of endocranial shape visually show that the developmental trajectories of *Alligator* and *Gallus* do not align with the principal evolutionary trajectory from ancestral shapes of non-avialan dinosaurs and crown birds (*Figures 2a* and *3a*). Comparison of these trajectories in full shape space (i.e., without the reduction in dimensionality of the data) further demonstrates that vectors of developmental shape change are different from the vector of ancestral endocranial shapes of coelurosaurs to crown birds (p<0.002). As such, paedomorphosis does not uniformly explain the major brain shape changes that occurred during the theropod-bird transition given our current sampling. This discordance with the results of previous studies on avian brain and skull evolution could be attributed to the exclusion of embryonic specimens that

encompass substantial changes in brain morphology. Embryonic specimens were not sampled in this study because endocasts cannot be reliably created for embryos with incomplete ossification, and merging endocast and brain shape data in archosaurs leads to an artifact where endocasts appear paedomorphic relative to corresponding brains (*Watanabe et al., 2019*). While their study was limited to analyzing 2D dorsal contour of endocasts, *Beyrand et al., 2019* have reported that during prenatal stages, crocodylians undergo a neuroanatomical transition from dorsoventrally flexed brain, akin to avian brains, to an anteroposteriorly linear brain profile. Therefore, at least some aspects of avian brain form are likely the result of paedomorphosis, a pattern which is not observed with postnatal developmental sampling.

Besides differences in developmental sampling, lack of uniform heterochronic signal in our data could be due to differences in (i) the morphological variation captured by high-density 3D compared to 2D data; (ii) the developmental trajectories of brain shape across archosaurs; and (iii) the impact of heterochronic processes across brain regions, where a mixture of forward and backward shifts in developmental processes occurred within localized regions, as has been shown in the cranium (*Bhullar et al., 2012*; *Plateau and Foth, 2020*). As mentioned above, a recent study investigating hetechrony as a mechanism in archosaurian brain evolution was restricted to analyzing the dorsal contour of the brain and endocasts in lateral view (*Beyrand et al., 2019*). This morphometric characterization precludes neuroanatomical variation within brain regions and along the mediolateral axis which account for 22.0% of our 3D dataset that may contribute a unique signal to endocranial shape variation. Another potential explanation for the absence of clear heterochronic signal is the intriguing possibility that non-avialan dinosaurs possessed a distinct modality in brain development from crocodylians and crown birds. Histological evidence suggests that IGM 100/1126 was an immature individual (*Erickson et al., 2007*) and its proximity to crown birds in endocast shape may signify that non-avialan dinosaurs had a unique postnatal development that begins with avian-like brain form (extrapolated section of the regression line for non-avialan dinosaurs in *Figure 3a*).

Lastly, regional analysis of neuroanatomical shape suggests that differing heterochronic signals within each brain region contribute to the lack of uniform heterochronic signal in our endocast data. Relative to the developmental trend in *Alligator* (decreasing PC1 residual shape scores in *Figure 3*), the cerebrum, optic lobe, and cerebellum trend toward increasingly paedomorphic shape (greater PC1 residual shape scores) starting from non-avialan dinosaurs and into modern birds (*Figure 3b–d*). In contrast, crown birds, including hatchling *Gallus*, exhibit derived, relatively more 'mature' medulla shapes beyond those of adult *Alligator*. Taken together, the piecemeal evolutionary assembly of the 'avian' brain may have comprised increasing paedomorphic effect on the cerebrum and optic lobe among non-avialan coelurosaurs and on the cerebellum among paravian dinosaurs, followed by peramorphosis of the medulla at least within crown birds. These mosaic patterns across taxa and anatomical regions only begin to exemplify the complexity of evolutionary and developmental interactions, illustrating how the identification of clear heterochronic signals is often more challenging and nuanced than implied by a single mechanism. As *Shea, 2002* proclaims regarding human morphology, 'there is no central component of heterochronic transformation that predominantly accounts for the bulk of morphogenetic and evolutionary transitions' (p. 95). We observe this pattern for coelurosaurian brain evolution as well.

## Crown birds possess a more integrated, not modular, brain structure

Brain evolution has proceeded through a mixture of integrated (concerted) (*Finlay and Darlington, 1995*; *Finlay et al., 2001*) and modular (mosaic) (*Barton and Harvey, 2000*; *Hager et al., 2012*; *Hoops et al., 2017*; *Sukhum et al., 2018*) patterns. Consistent with volumetric studies on avialan and non-avialan coelurosaurian neuroanatomical studies (*Iwaniuk et al., 2004*; *Balanoff et al., 2016b*; *Sayol et al., 2016a*; *Moore and DeVoogd, 2017*), our study points to both integrated and modular trends shaping their brain evolution via common allometric and regional trends. In humans and chimpanzees, the highly encephalized brain is thought to have coincided with the acquisition of a more modular structure than other mammals (*Gómez-Robles et al., 2014*). This scenario agrees with the traditional notion that a more modular structure, allowing for quasi-independent changes among regions, promotes the evolution of novel and diverse forms (*Wagner, 1996*; *Wagner and Altenberg, 1996*; *Klingenberg, 2005*). Interestingly, we find that the brains of crown birds are more integrated than those of non-avialan archosaurs, and this difference is broadly reflected, albeit weakly, in the postnatal development of *Gallus* and *Alligator*. This outcome is particularly surprising

given that the allometric signal, a strong integrating factor, is greater in non-avialan coelurosaurs than extant birds (*Supplementary file 1c*). The avian brain, therefore, counters the notion that structures become increasingly modular through macroevolutionary time to maintain or increase evolvability (*Wagner and Altenberg, 1996*). Although seemingly counter-intuitive, recent empirical and simulation studies demonstrate that integrated structures have the capacity to evolve more extreme phenotypes when selection acts along major axes of variation (*Villmoare, 2013*; *Goswami et al., 2015*; *Felice et al., 2018*; *Machado et al., 2018*; *Rolian, 2019*). As such, the neuroanatomical diversity observed across Neornithes could still arise from strongly integrated brain structure. This result aligns with a recent large-scale analysis on volumetric data showing that crown birds possess greater brain-to-body size integration relative to non-avialan dinosaurs (*Ksepka et al., 2020*). Therefore, a more integrated structure seems to underlie brain shape and size evolution within Neornithes relative to their coelurosaurian ancestors.

Whether the evolution of a highly encephalized brain with inflated cerebra emerged from an ancestrally more modular or the derived, more integrated configuration remains to be examined with additional endocasts from basally divergent members of Avialae. Volumetric evidence indicating pulses of cerebral expansion occurring among non-avialan maniraptoran dinosaurs (*Balanoff et al., 2013*; *Ksepka et al., 2020*) suggests that an ancestrally more modular brain allowed for increasingly encephalized brains and globular cerebra to evolve prior to, and even after, the origin of Avialae. If true, the more integrated brain of crown birds may be a consequence of the subsequent 'spatial packing' of brain tissue inside the endocranial cavity, a hypothesis proposed for the flexed profile of some mammalian brains (*Lieberman et al., 2008*). A more integrated brain could also be attributed to the abbreviated growth period in crown birds which could be reducing the cumulative imprinting of new covariation patterns onto the integration pattern established earlier in development (*Hallgrímsson et al., 2009*; *Goswami et al., 2014*).

Besides the overall strength of integration, the degree of correlation between regions helps formulate mechanistic explanations based on the premise that strongly integrated regions are thought to emerge through shared spatial, functional, developmental, and genetic factors (*Wagner and Altenberg, 1996*; *Klingenberg, 2008*; *Gómez-Robles et al., 2014*). For example, previous studies have shown that the strength of axonal connections in the brain is associated with the extent of cortical surface folds in mammals (*Hofman, 2014*). Although the link between surface morphology and neuronal connections in avian systems is yet unclear (although see *Early et al., 2020*), strongly correlated shape changes could also represent functional coordination between regions. Vision is the dominant sensory modality in modern birds (*Shimizu et al., 2010*; *Walsh and Milner, 2011*; *Martin, 2014*), and their visual pathways include major projections from the optic lobe to the cerebrum, including the Wulst (*Wylie et al., 2009*; *Shimizu et al., 2010*). These critical neuronal connections may induce coordinated morphological development and evolution between the cerebrum and optic lobe shapes. Interestingly, non-avialan coelurosaurs exhibit the strongest integration between the cerebrum and optic lobe and within the hindbrain (cerebellum and medulla). The presence of strong integration between the cerebrum and optic lobe in non-avialan coelurosaurs, but not in a developing *Alligator*, is consistent with the inference from allometric trajectories that derived non-avialan coelurosaurs already possessed aspects of the avian-grade cerebrum and optic lobes. Collectively, these results suggest that key aspects of the 'avian' visual system emerged in non-avialan dinosaurs, preceding the origin of birds and powered flight. Visual acuity, perhaps for predation or signaling through colorful feathers, was likely an important facet of their lifestyle, an evolutionary scenario shared with primate brain evolution (*Barton, 1998*; *Kirk, 2006*).

## Materials and methods

### Specimens

CT data and endocranial reconstructions were sampled from previously published studies (*Balanoff et al., 2013*; *Gold and Watanabe, 2018*; *Watanabe et al., 2019*). The interspecific dataset includes six non-avialan coelurosaurs, 37 neornithines, and *Archaeopteryx*. Among non-avialan dinosaurs, we sampled coelurosaurs due to their phylogenetic affinity to birds, and crucially, major neuroanatomical regions are visible on their endocasts unlike those of more basally diverging theropods (*Paulina-Carabajal et al., 2019*). The braincases of *Alioramus* and *Incisivosaurus* are

taphonomically deformed which would lead to inaccurate characterization of endocranial shape. Because the endocast of *Alioramus* showed approximately uniform shear, the endocranial model and coordinate data were retrodeformed based on the symmetrization algorithm (*Ghosh et al., 2010*) implemented in the Morpho R package based on discrete landmarks that are bilaterally symmetric (*Schlager et al., 2018*). In contrast, the endocast of *Incisivosaurus* shows mediolateral compression (*Balanoff et al., 2009*) which impedes reliable retrodeformation with existing tools. However, *Incisivosaurus* occupies regions of morphospaces that are compatible with other non-avian coelurosaurs, with the exception of cerebrum shape (*Figure 3b*). Statistical analyses without *Incisivosaurus* generate results that are consistent with those presented here, including crown birds possessing significantly more integrated brain architecture (partial least squares [PLS] effect size difference = 3.911; p<0.001). Therefore, we have presented the results which include endocranial shape data from *Incisivosaurus*.

Rockefeller Wildlife Refuge (Grand Chenier, LA) provided individuals of *A. mississippiensis*: five <1 year, two 1–2 years, and two 2–3 years of age (n = 11). Older alligator specimens were collected by a nuisance trapper (Vaughan Gators, Tallahassee, FL). Charles River Laboratories (North Franklin, CT) supplied male chicken specimens (*G. gallus*) at 1 day, 1 week, 3 weeks, 6 weeks, and >8 weeks of age. Two individuals were sampled for each age group, with the exception of four individuals at 1 day and >8 weeks of age (n = 14). We selected *Gallus* as exemplar taxon for crown birds due to the availability of developmental series and their importance as a model system. The alligator and chicken specimens were submerged in 10% neutral-buffered formalin immediately following euthanasia (Stony Brook IACUC Protocol #236370–1, Oklahoma State University Center for Health Sciences IACUC Protocol #2015–1 for alligator specimens; chicken specimens were euthanized by Charles River Laboratories). These specimens were fixed in formalin for over 8 weeks before imaging to mitigate soft-tissue distortions (*Weisbecker, 2012*). Please refer to *Watanabe et al., 2019* for additional details on sampling and imaging of *Alligator* and *Gallus* specimens.

## Imaging and endocranial reconstructions

The heads of specimens were CT-scanned at multiple institutions using variable scan parameter values to optimize the contrast and resolution of the X-ray images, while also considering available scan time. For larger specimens requiring multiple scans, separate image stacks were fused using the '3D Stitching' function in ImageJ (FIJI) v1.49u (*Schindelin et al., 2012*). In VGStudio MAX v2.2 (Volume Graphics, Heidelberg, Germany), full image stacks of each specimen were imported, and virtual segmentation was conducted following the protocol outlined by *Balanoff et al., 2016a*. Reconstructed endocasts were then exported as 3D polygon mesh files. Based on the same landmark scheme analyzed in this study, endocasts are known to accurately represent the variation in brain size and shape in archosaurs and follow the same ontogenetic trends as brain shape in *Alligator* and *Gallus* (*Watanabe et al., 2019*). As such, we considered the directionality and the variance-covariance structure of brain shape to be closely reflected by endocranial shape data given the large-scale comparative sampling of our study.

## Morphometric data

We employed a high-density 3D landmark-based GM approach to characterize the shape and size of endocasts and their major functional regions (*Figure 1*; *Supplementary file 1b*). The landmark scheme combines discrete landmarks with semi-landmarks on curves and surfaces using Landmark Editor v3.6 (*Wiley et al., 2005*). Its 'patch' tool allows the placement of discrete, consistently defined landmarks at junction points of major brain regions (i.e., left and right cerebral hemispheres, optic tecta, cerebellum, medulla) and specified density of semi-landmarks within these partitions (see *Supplementary file 1b*). Despite its critical role in the neurosensory repertoire, we did not characterize the shape of the olfactory tract and bulbs due to the incomplete preservation of this region in fossil taxa. To extract shape data, we subjected the coordinate data to a generalized Procrustes alignment (*Gower, 1975*; *Rohlf and Slice, 1990*) minimizing total bending energy, while allowing semi-landmarks to slide on the mesh surface (*Gunz et al., 2005*; *Gunz and Mitteroecker, 2013*). This was achieved using the slider3d and gpagen functions in the R packages Morpho v2.7 (*Schlager, 2017*) and geomorph v3.2.1 (*Adams and Otárola-Castillo, 2013*), respectively. To remove redundant shape information but also avoid artifacts from aligning one side of bilaterally

symmetric structures (*Cardini, 2016*; *Cardini, 2017*), right landmarks were removed after aligning bilateral coordinate data (*Bardua et al., 2019*). Ultimately, the left and midline landmarks were analyzed, including the left cerebrum (54 landmarks), left optic lobe (29 landmarks), left side of cerebellum (18 landmarks), and left side of medulla (18 landmarks).

We generated two versions of the regional shape datasets—one based on global alignment of entire endocranial data and second based on local alignment of regional shape data. The former captures variation in both regional shape and relative position within the endocast, whereas the latter dataset exclusively characterizes the intrinsic shape of each region. We primarily report results based on locally aligned regional shape data to mitigate the effect of relative positions of each region on the coordinate data which would inflate the magnitude of integration between regions, as well as shape differences (e.g., optic lobe located posterior to the cerebrum in *Alligator* and posteroventral to the cerebrum in crown birds). Results based on globally aligned regional shape data, along with locally aligned data, are presented in the supplementary information (*Supplementary file 1c–f*; *Figure 4—figure supplement 1*). Besides shape, log-transformed centroid size of endocasts was calculated from the coordinate data, which are known to be a reliable proxy for brain and body size across birds and alligators (*Marugán-Lobón et al., 2016*; *Watanabe et al., 2019*). We assessed the relative magnitude of digitization error by repeatedly collecting landmark data from a 1-day-old chicken (10 replications), which accounted for 2.41% of the total shape variation of the dataset and was thus considered to be negligible.

## Time-calibrated phylogeny

First, we created a maximum clade credibility tree of extant birds from 3000 posterior trees based on Hackett tree backbone available on birdtree.org (*Jetz et al., 2012*) using TreeAnnotator v1.8.1 (*Drummond et al., 2012*). *Apteryx* sp., *Diomede* sp., and *Eudyptes* sp. in our sampling were treated as *A. australis*, *D. exulans*, and *E. chrysocome* for the purpose of constructing a tree including all sampled taxa in this study. Then, we incorporated *Alligator*, *Archaeopteryx*, and non-avialan dinosaurs to the tree based on the mean age of first occurrence age listed in the Paleobiology Database (paleobiodb.org). Ages of internal nodes were determined by the maximum age between sister groups to which the species belong (e.g., age of Paraves determined by maximum age of Deinonychosauria and Avialae). When the maximum age of sampled specimen was identical to that of its clade, the age of the internal node was set to equally bisect the parent and descendent branch (*Bell and Lloyd, 2015*). The Dodo (*Raphus cucullatus*) was placed based on estimated divergence from *Caloenas* lineage at 15.1 Ma (*Pereira et al., 2007*). Similarly, the Great Auk (*Pinguinus impennis*) was placed based on the mean stratigraphic age of earliest occurrence of its sister group *Alca* (*Smith, 2015*). This combined paleontological and neontological tree was then modified to reflect the updated topology and branch lengths proposed by a recent genomic study (*Prum et al., 2015*). For sampled species not included in the genomic tree, a closest relative was chosen based on the global tree of birds (*Jetz et al., 2012*).

## Analysis

All statistical analyses were performed in R version 4.0.3 (*R Development Core Team, 2020*). To visualize patterns of neuroanatomical variation, morphospaces for endocasts and their regions (cerebrum, optic lobe, cerebellum, medulla) were constructed using scores from PCA on shape data. The degree of phylogenetic signal, allometry, and evolutionary allometry was assessed with the physignal, procD.lm, and procD.pgls functions, respectively, in the geomorph package with 1000 pseudo-replications. These multivariate statistical tests have been demonstrated to be robust against type I error and loss of power associated with specimen and landmark sampling (*Adams, 2014a*; *Adams, 2014b*; *Collyer et al., 2015*). For visualizing allometric trends, we plotted the PC1 of residuals from the overall shape-to-size relationship against scores along this allometric relationship (*Mitteroecker et al., 2004*). The CAC function in the Morpho package was used to extract CAC scores and residuals from this trend. Statistical differences between endocranial shapes and allometric trajectories between clades were tested with the procD.lm function. We used the angleTest function in the Morpho R package to test for differences between vectors of evolutionary and developmental shape change in full shape space. The evolutionary shape vector was created from ancestral shape reconstruction for Coelurosauria and Neornithes using the anc.recon function in the

Rphylopars package (*Goolsby, 2016*). Developmental shape vectors were formulated using smallest and largest endocasts sampled for *Alligator* and *Gallus*. Finally, we used two different statistics to measure the degree of integration among the brain regions—rho based on ML (*Goswami and Finarelli, 2016*) and covariance ratio using the modularity.test function (*Adams, 2016*). Although known to be susceptible to specimen and landmark sampling (*Adams and Collyer, 2016*), results based on correlation coefficients from partial least squares ($R_{PLS}$) using integration.test are also presented in the supplementary information (*Supplementary file 1d, e*). Tests of neuroanatomical integration on crown birds excluded *Gallus* to maintain separation from its developmental analysis. To test for one-tailed differences in the degree of integration between clades, we used the compare.pls function in the geomorph package which is robust to differences in specimen and landmark sampling (*Adams and Collyer, 2016*). For statistical tests of interspecific data, we corrected the shape data for phylogenetic structure based on phylogenetic generalized least-squares method with the exception of allometric trajectory comparison between non-avalian dinosaurs and crown birds.

## Data availability

Aligned bilateral landmark data with original size information intact are available on Dryad (doi:10.5061/dryad.qv9s4mwdk). The R code to perform the analyses presented in the study is accessible on Github (https://github.com/akiopteryx/analyses; copy archived at swh:1:rev:c1df76ef1c770-d464a9e8def31a763c8a47e58ba; *Watanabe, 2021*).

## Acknowledgements

We are indebted to Ruth Elsey (Rockefeller Wildlife Refuge), Gregory Erickson (Florida State University), David Kay (currently Oklahoma State University, Center for Health Sciences), Broderick Vaughan (Vaughan Gators), and Doug Warner (Charles River Laboratories) for providing *Alligator* and *Gallus* specimens; Morgan Hill Chase (American Museum of Natural History) and Henry Towbin (currently Columbia University), as well as Johnny Ng and Cheuk Ying Tang (Mount Sinai Hospital), for assistance with CT imaging; Isabelle Brenes and Carolynn Merrill for assistance with processing and segmenting CT data; William Harcourt-Smith, Emma Sherratt, and Jesus Marugán-Lobón for helpful discussions related to this study; and Min Zhu, Alice Clement, and Federico Degrange for constructive and thoughtful review of the manuscript.

## Additional information

### Funding

| Funder | Grant reference number | Author |
|---|---|---|
| National Science Foundation | Graduate Research Fellowship | Akinobu Watanabe |
| National Science Foundation | DEB-1406849 | Akinobu Watanabe |
| National Science Foundation | DEB-1311790 | M Eugenia L Gold |
| National Science Foundation | DEB-1801224 | Amy M Balanoff Mark A Norell |
| National Science Foundation | DEB-1457180 | Paul M Gignac |
| National Science Foundation | DEB-1754659 | Paul M Gignac |
| Society of Vertebrate Paleontology | Mary R. Dawson Predoctoral Fellowship Grant | Akinobu Watanabe |
| Jurassic Foundation | | Akinobu Watanabe |
| Macaulay Family Endowment | | Mark A Norell |
| Newt and Callista Gingrich Endowment | | Mark A Norell |
| American Museum of Natural History | Division of Paleontology | Akinobu Watanabe Amy M Balanoff Paul M Gignac |

M Eugenia L Gold
Mark A Norell

The funders had no role in study design, data collection and interpretation, or the decision to submit the work for publication.

## Author contributions

Akinobu Watanabe, Conceptualization, Resources, Data curation, Formal analysis, Funding acquisition, Validation, Investigation, Visualization, Methodology, Writing - original draft, Project administration, Writing - review and editing; Amy M Balanoff, Paul M Gignac, M Eugenia L Gold, Resources, Funding acquisition, Writing - review and editing; Mark A Norell, Resources, Data curation, Supervision, Funding acquisition, Writing - review and editing

## Author ORCIDs

Akinobu Watanabe https://orcid.org/0000-0001-5057-4772
Amy M Balanoff https://orcid.org/0000-0003-4030-3818
Paul M Gignac http://orcid.org/0000-0001-9181-3258
M Eugenia L Gold https://orcid.org/0000-0003-4317-3625

## Ethics

Animal experimentation: The handling of alligator specimens used in this study was approved by institutional animal care and use committee (IACUC) where the protocols were conducted (Stony Brook University IACUC protocol #236370-1, Oklahoma State University Center for Health Sciences IACUC protocol #2015-1). The chicken specimens were euthanized by Charles River Laboraties. The chicken and alligator specimens were euthanized using cervical dislocation and administration of pentobarbital sodium, respectively, and every effort was made to minimize suffering.

## Decision letter and Author response

Decision letter https://doi.org/10.7554/eLife.68809.sa1
Author response https://doi.org/10.7554/eLife.68809.sa2

# Additional files

## Supplementary files

• Supplementary file 1. Supplementary tables related to this study. (a) List of taxa sampled for this study, with the exclusion of *Alligator* and *Gallus*. Institutional abbreviations: AMNH, American Museum of Natural History, New York, NY, USA; BMNH, British Museum of Natural History, London, UK; FMNH, Field Museum of Natural History, Chicago, IL, USA; KU, University of Kansas, Lawrence, KS, USA; NMNH, National Museum of Natural History, Washington DC, USA; TCWC, Texas Cooperative Wildlife Collection, College Station, TX, USA; TMM, Texas Memorial Museum, Austin, TX, USA; WDC, Wyoming Dinosaur Center, Thermopolis, WY, USA. (b) List of discrete landmarks and density of semi-landmarks for each neuroanatomical region. (c) Phylogenetic signal (Blomberg's *K*), allometry, and evolutionary allometry in endocranial shape. Results generated using physignal, procD.lm, procD.pgls functions in geomorph R package v3.2.1 (*Adams and Otárola-Castillo, 2013*). Results from analysis on globally and locally aligned regions are presented as first and second values within a cell, respectively. Allometry evaluated with log-transformed centroid size of the entire endocast and local region, respectively. *, **, and *** indicate $p < 0.05$, $< 0.01$, and $< 0.001$, respectively, based on 1000 pseudo-replications. (d) Integration within and between locally aligned neuroanatomical regions. Degree of integration is measured by correlation coefficient from two-block partial least-squares analysis ($R_{PLS}$; upper off-diagonal) and correlation coefficient ($\rho$; diagonal, lower off-diagonal) using the R packages geomorph v3.2.1 (*Adams and Otárola-Castillo, 2013*) and EMMLiv2 v0.0.3 (*Goswami and Finarelli, 2016*), respectively. Interspecific analyses are phylogenetically corrected using phylogenetic generalized least-squares method. (e) Integration within and among globally aligned neuroanatomical regions. The degree of integration is measured by correlation coefficient from two-block partial least-squares analysis ($R_{PLS}$; upper off-diagonal) and correlation

coefficient (ρ; diagonal, lower off-diagonal) using the R packages geomorph v3.2.1 (*Adams and Otárola-Castillo, 2013*) and EMMLi v2 v0.0.3 (*Goswami and Finarelli, 2016*), respectively. Interspecific analyses are phylogenetically corrected using phylogenetic generalized least-squares method. (f) Integration between neuroanatomical regions using covariance ratios (CR) (*Adams, 2016*). Degree of integration between globally aligned regional shapes are listed in the upper off-diagonal elements and that of locally aligned regional shapes in the lower off-diagonal elements. Interspecific analyses are phylogenetically corrected using phylogenetic generalized least-squares method. (g) Comparison of integration among neuroanatomical regions using the compare.pls function in the geomorph R package (*Adams and Otárola-Castillo, 2013*; *Adams and Collyer, 2016*). '+' and '−' denote greater and lesser integration in Neornithes and *Gallus* compared to non-avialan coelurosaurs and *Alligator*, respectively. Integration among species calculated upon phylogenetic correction. *, **, and *** indicate p<0.05, <0.01, and <0.001 (one-tailed), respectively, based on 1000 pseudo-replications. Numbers preceding and following '/' indicate results based on globally and locally aligned data, respectively.

- Transparent reporting form

## Data availability

Landmark data are included in the supporting files and also deposited in Dryad (https://doi.org/10.5061/dryad.qv9s4mwdk). In addition, we have uploaded the R code to perform analyses presented in the study to Github: https://github.com/akiopteryx/analyses (copy archived at https://archive.softwareheritage.org/swh:1:rev:c1df76ef1c770d464a9e8def31a763c8a47e58ba).

The following dataset was generated:

| Author(s) | Year | Dataset title | Dataset URL | Database and Identifier |
|---|---|---|---|---|
| Watanabe A, Balanoff AM, Gignac PM, Gold MEL, Norell MA | 2021 | High-density 3-D coordinate data of avian and non-avian dinosaur endocasts | https://doi.org/10.5061/dryad.qv9s4mwdk | Dryad Digital Repository, 10.5061/dryad.qv9s4mwdk |

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
