## [Decision Letter]

**Acceptance summary:**

This manuscript is of broad interest to vertebrate paleontologists, evolutionary and developmental biologists, and neuroanatomists. It provides valuable 3-D imaging, high-density shape data of endocasts of crown birds and non-avialan coelurosaurian dinosaurs, on which patterns of covariation in archosaur brain anatomy are analysed.

**Decision letter after peer review:**

Thank you for submitting your article "Novel neuroanatomical integration and scaling define avian brain shape evolution and development" for consideration by *eLife*. Your article has been reviewed by 3 peer reviewers, including Min Zhu as the Reviewing Editor and Reviewer #1, and the evaluation has been overseen by George Perry as the Senior Editor. The following individuals involved in review of your submission have agreed to reveal their identity: Alice Clement (Reviewer #2); Federico Degrange (Reviewer #3).

For the revisions, see the comments below from the three reviewers.

*Reviewer #1 (Recommendations for the authors):*

I have the following comments for your consideration.

l.20-21: Add Archaeopteryx, which is a stem bird but not a non-avialan coelurosaurian dinosaur.

l.25: Use 'avian'-grade or avian-grade consistently throughout the text.

l.66: "truly unique" is puzzling at the first glance. "derived" might be better here.

l.99: Delete "i.e., excluding Archaeopteryx" , which is logically inconsistent with the following note "Archaeopteryx, often considered one of the earliest diverging avialan".

l.235: "theropod-bird transition" is more accurate.

l.253: "…the avian skull (Bhullar et al., 2012)-a structure physically, functionally, and developmentally linked to the brain…" looks redundant.

l.308: "through deep time" is puzzling here. Delete?

*Reviewer #2 (Recommendations for the authors):*

A great paper! I have very little to add.

• Line 58-59: Surely the following reference is also relevant here:

Early CM, Iwaniuk AN, Ridgely RC, and Witmer LM. 2020. Endocast structures are reliable proxies for the sizes of corresponding regions of the brain in extant birds. Journal of Anatomy:1-15.

• In addition to modularity discussed here, perhaps worthy to note (either ~ line 300 or elsewhere in discussion) recent work such as Smaers et al. (2021) discussing allometric shifts in mammalian brain evolution resulting from either brain or body size changes:

Smaers JB, et al. 2021. The evolution of mammalian brain size. Science Advances 7.

• Interpretation of Figure 4 might be improved by including a colour-coded endocast (as per Figure 1) showing e.g. Cr = cerebrum, Me = medulla etc. Also taxon names are TINY and very hard to read.

*Reviewer #3 (Recommendations for the authors):*

I really enjoyed reading this manuscript and I congratulate (and thank) the authors for a well written research. Some issues to be fixed are as follows:

1) Line 106. It would be good to describe here what positive and negative values for the illustrated PC1 and PC2 mean (e.g., positive values for PC1 are related to less expended cerebrum and a less flexed brain, etc, etc)

2) lines 114 and 115. Should be 2b, c and 2d,e.

3) lines 129-131. Take into account that although the statement is true (for the morphospace), it will be better to be more careful about this statement. Since the troodontid is below the trajectory line, it is not quite correct to state that it is actually between Neornithes and non-avialan dinosaurs.

4) line 143. In figure 3b, the troodontid is not indicated. Is there a reason for this?

5) Remove b from Balanoff et al., 2016 (lines 65, 207, 297).

6) line 243. This hould be Sayol et al., 2016a. Line 297. This should be Sayol et al., 2016b

7) line 249..…(i.e., retention.…

8) line 249. replace brain morphology by features.

9) lines 265-266. As you discussed (ii) and (iii), I think that (i) has to be also discussed somehow. Are there any other studies of 2 or 3D geomorphometrics of the brain that show similar/different results? How? Also, if there are, comments about their strengths and flaws need the be mentioned.

10) Line 335 and 380. "et al." without italics.

11) Line 337. I suggest to add some reference of Martin, such as: Martin (2007) Visual fields and their functions in birds, and Martin (2014) The subtlety of simple eyes: the tuning of visual fields to perceptual challenges in birds

---

## [Author Response]

Reviewer #1 (Recommendations for the authors):I have the following comments for your consideration.l.20-21: Add Archaeopteryx, which is a stem bird but not a non-avialan coelurosaurian dinosaur.l.25: Use 'avian'-grade or avian-grade consistently throughout the text.l.66: "truly unique" is puzzling at the first glance. "derived" might be better here.l.99: Delete "i.e., excluding Archaeopteryx" , which is logically inconsistent with the following note "Archaeopteryx, often considered one of the earliest diverging avialan".l.235: "theropod-bird transition" is more accurate.l.253: "…the avian skull (Bhullar et al., 2012)-a structure physically, functionally, and developmentally linked to the brain…" looks redundant.l.308: "through deep time" is puzzling here. Delete?

These suggestions by Reviewer #1 have been addressed in the revised manuscript. For the comment related to Line 253, we have replaced “physically” with “topologically” to be more specific about what we meant by “physical” relationship of the brain and the skull. For the comment related to Line 308, we have replaced “through deep time” with “through macroevolutionary time” to be more precise in our language.

Reviewer #2 (Recommendations for the authors):A great paper! I have very little to add.• Line 58-59: Surely the following reference is also relevant here:Early CM, Iwaniuk AN, Ridgely RC, and Witmer LM. 2020. Endocast structures are reliable proxies for the sizes of corresponding regions of the brain in extant birds. Journal of Anatomy:1-15.

We agree with the reviewer that this reference should be cited here, in addition to being cited in the Discussion section. We have added the suggested reference.

• In addition to modularity discussed here, perhaps worthy to note (either ~ line 300 or elsewhere in discussion) recent work such as Smaers et al. (2021) discussing allometric shifts in mammalian brain evolution resulting from either brain or body size changes:Smaers JB, et al. 2021. The evolution of mammalian brain size. Science Advances 7.

Although we initially considered referencing the study by Smaers and colleagues, we have decided to not incorporate it into the text as suggested because these studies concern the integration between brain and body size, whereas our study investigates the degree of integration within the brain. While I think there may be a way to include the work by Smaers et al. and other similar studies on brain-body integration in this part of the manuscript, I am also concerned that this may invite criticism or confusion because the anatomical scope and topic of the study are different enough from our study.

• Interpretation of Figure 4 might be improved by including a colour-coded endocast (as per Figure 1) showing e.g. Cr = cerebrum, Me = medulla etc. Also taxon names are TINY and very hard to read.

Thank you for the suggestion. When creating Figure 4, I (A.W.) initially tried to fit the full names (e.g., “cerebrum” instead of “Cr”) into the circles but was unable to without making the labels too small for legibility. I also attempted to include a diagram of an endocast similar to what is suggested but found that it makes the figure appear too busy in my view. The Figure 4 in its current state represents what I believe to be an adequate compromise, with color-coded circles that match the color scheme of the endocast in Figure 1. Meanwhile, we have enlarged the tip labels on the phylogenetic tree in Figure 1 to increase legibility as suggested by the reviewer.

Reviewer #3 (Recommendations for the authors):I really enjoyed reading this manuscript and I congratulate (and thank) the authors for a well written research. Some issues to be fixed are as follows:1) Line 106. It would be good to describe here what positive and negative values for the illustrated PC1 and PC2 mean (e.g., positive values for PC1 are related to less expended cerebrum and a less flexed brain, etc, etc)

We appreciate the reviewer’s suggestion. The reason why I (A.W.) typically do not report shape changes associated positive and negative values of PC scores in my manuscripts is because the positive-negative signs of PC scores is purely arbitrary and can be switched. However, I also understand that this information may help orient the readers when they are examining the morphospaces so we have added a description of these shape changes. In addition, we did not include shape changes associated PC2 in our manuscript, so we have also included this information in the revised manuscript.

2) lines 114 and 115. Should be 2b, c and 2d,e.

Thank you for catching this careless mistake.

3) lines 129-131. Take into account that although the statement is true (for the morphospace), it will be better to be more careful about this statement. Since the troodontid is below the trajectory line, it is not quite correct to state that it is actually between Neornithes and non-avialan dinosaurs.

Thank you for the suggestion. We have modified the statements to focus on *Archaeopteryx* and excluded the statement referring to the troondontid as an intermediary taxon.

4) line 143. In figure 3b, the troodontid is not indicated. Is there a reason for this?

The unnamed troondontid is labeled in Figure 3b. Therefore, it’s unclear to me what the reviewer is referring to with this comment. As such, we did not alter Figure 3b.

5) Remove b from Balanoff et al., 2016 (lines 65, 207, 297).

Thank you for pointing out the fact that “Balanoff et al., 2016a” was not specified in the manuscript. We have kept Balanoff et al., 2016b, but have added “2016a” in Ln 458.

6) line 243. This should be Sayol et al., 2016a. Line 297. This should be Sayol et al., 2016b.

The letter assignment is based on the order listed in the References section, not in the order of appearance in the main text. If this is pointed out by an *eLife* editor, then we will make sure to re-order these two references.

7) line 249..…(i.e., retention.…

We think that “i.e.” is not necessary here so we have kept the original text here.

8) line 249. replace brain morphology by features.

We think that “morphology” is appropriate here, so we have kept the original wording.

9) lines 265-266. As you discussed (ii) and (iii), I think that (i) has to be also discussed somehow. Are there any other studies of 2 or 3D geomorphometrics of the brain that show similar/different results? How? Also, if there are, comments about their strengths and flaws need the be mentioned.

We have added a note and brief discussion on Beyrand et al. 2019, which is a recent study that used 2-D morphometric methods to study potential heterochronic shifts in archosaurian brain evolution. We also added our developmental sampling as to why our dataset does not exhibit clear and broad heterochronic signal as in this previous study.

10) Line 335 and 380. "et al." without italics.

This has been fixed.

11) Line 337. I suggest to add some reference of Martin, such as: Martin (2007) Visual fields and their functions in birds, and Martin (2014) The subtlety of simple eyes: the tuning of visual fields to perceptual challenges in birds.

Thank you for informing us about these articles. In my view, the first study (Martin 2007) examines variation in visual field and does not focus on the high visual acuity and need in birds. The second study (Martin 2014) does highlight the importance of vision in birds, so I have decided to cite this article.